**Subject Category:**
Biology (whole organism)

ecology/environmental science/microbiology

disease-suppressive soil, *glycine max*, *meloidogyne*, nematophagous fungi, phytopathogenic pathogens and pests, sustainable agriculture

**Author for correspondence:**
Hirokazu Toju
e-mail: toju.hirokazu.4c@kyoto-u.ac.jp

# Consortia of anti-nematode fungi and bacteria in the rhizosphere of soybean plants attacked by root-knot nematodes

## Hirokazu Toju[1,2] and Yu Tanaka[2,3]

[1]Center for Ecological Research, Kyoto University, Otsu, Shiga 520-2133, Japan
[2]Precursory Research for Embryonic Science and Technology (PRESTO), Japan Science and Technology Agency, Kawaguchi, Saitama 332-0012, Japan
[3]Graduate School of Agriculture, Kyoto University, Kitashirakawa-oiwake-cho, Sakyo, Kyoto 606-8502, Japan

HT, 0000-0002-3362-3285

Cyst and root-knot nematodes are major risk factors of agroecosystem management, often causing devastating impacts on crop production. The use of microbes that parasitize or prey on nematodes has been considered as a promising approach for suppressing phytopathogenic nematode populations. However, effects and persistence of those biological control agents often vary substantially depending on regions, soil characteristics and agricultural practices: more insights into microbial community processes are required to develop reproducible control of nematode populations. By performing high-throughput sequencing profiling of bacteria and fungi, we examined how root and soil microbiomes differ between benign and nematode-infected plant individuals in a soybean field in Japan. Results indicated that various taxonomic groups of bacteria and fungi occurred preferentially on the soybean individuals infected by root-knot nematodes or those uninfected by nematodes. Based on a network analysis of potential microbe–microbe associations, we further found that several fungal taxa potentially preying on nematodes (*Dactylellina* (Orbiliales), *Rhizophydium* (Rhizophydiales), *Clonostachys* (Hypocreales), *Pochonia* (Hypocreales) and *Purpureocillium* (Hypocreales)) co-occurred in the soybean rhizosphere at a small spatial scale. This study suggests how 'consortia' of anti-nematode microbes can derive from indigenous (resident) microbiomes, providing basic information for managing anti-nematode microbial communities in agroecosystems.

# 1. Introduction

Plant pathogenic nematodes, such as cyst and root-knot nematodes, are major threats to crop production worldwide [1,2]. Soybean fields, in particular, are often damaged by such phytopathogenic nematodes, resulting in substantial yield loss [3,4]. A number of chemical nematicides and biological control agents (e.g. nematophagous fungi in the genera *Purpureocillium* and *Clonostachys*) have been used to suppress nematode populations in farmlands [5,6]. However, once cyst and root-knot nematodes appear in a farmland, they often persist in the soil for a long time [7], causing high financial costs in agricultural management. Finding ways to suppress pathogenic nematode populations in agroecosystems is a key to reducing risk and management costs in production of soybean and other crop plants.

To reduce damage by cyst and root-knot nematodes, a number of studies have evaluated effects of crop varieties/species, crop rotations, fertilizer inputs and tillage intensity on nematode density in farmland soil [1,8–10]. However, the results of those studies varied considerably depending on regions, soil characteristics and complicated interactions among multiple factors (e.g. interactions between organic matter inputs and tillage frequency) [11]. Thus, it remains an important challenge to understand the mechanisms by which phytopathogenic nematode populations are suppressed in some farmland soils but not in others [12]. New lines of information are required for building general schemes for making agroecosystems robust against the emergence of pest nematodes.

Based on the technological advances in high-throughput DNA sequencing, more and more studies have examined structures of microbial communities (microbiomes) in order to evaluate biotic environmental conditions in the endosphere and rhizosphere of plants [13–16]. Recent studies have uncovered microbiome compositions of 'disease-suppressive soils', in which pests and pathogens damaging crop plants have been suppressed for long periods of time [17–19]. Some studies have further discussed how some microbes within such disease-suppressive microbiomes contribute to the health and growth of crop plant species [17,20,21]. In one of the studies, soil microbiome compositions were compared among soybean fields that differed in the density of cyst nematodes [12]. The study then revealed that bacteria and fungi potentially having negative impacts on nematode populations (e.g. *Purpureocillium* and *Pochonia*) were more abundant in the long term than in short-term monoculture fields of soybeans [12]. Such among-farmland comparisons have provided invaluable insights into ecosystem functions of indigenous (native) microbiomes. Nonetheless, the potential relationship between cropping system management and community processes of anti-nematode microbes remains obscured because the farmlands compared in those studies could vary in climatic and edaphic factors. Moreover, because incidence of cyst and root-knot nematodes generally varies at small spatial scales [22], there can be spatial heterogeneity in abundance and community compositions of anti-nematode bacteria and fungi within a farmland. Studies focusing on fine-scale assembly of anti-nematode microbes are required for developing agroecosystem management protocols for controlling phytopathogenic nematodes.

We conducted an Illumina sequencing analysis of bacteria and fungi in a soybean (*Glycine max*) field and then examined how root and rhizosphere microbiome structures varied among host plant individuals that differed in damage by root-knot nematodes (*Meloidogyne* sp.). Based on the data of microbiomes at a small spatial scale, we statistically explored microbial species/taxa that occurred preferentially in the roots or rhizosphere soil of nematode-infected soybean individuals. We further investigated the structure of networks depicting co-abundance patterns of microbial species/taxa within the soybean field, thereby examining whether multiple anti-nematode bacteria and fungi form consortia (assemblages) on/around the plant individuals infected by root-knot nematodes. Our results suggest that various taxonomic groups of anti-nematode bacteria and fungi are present within indigenous microbiomes. This study also suggests that microbiome assembly at fine spatial scales is a key to managing populations and communities of such functional microbes.

# 2. Methods

## 2.1. Sampling

Fieldwork was conducted at the soybean field on the Hokubu Campus of Kyoto University, Japan (35.033°N, 135.784°E). In the field, the soybean strain 'Sachiyutaka' was sown at 15 cm intervals in two lines (electronic supplementary material, figure S1) on 4 July 2016 (basal fertilizer, N : P$_2$O$_5$ : K$_2$O = 3 : 10 : 10 g m$^{-2}$). In the lines, 69 and 62 individuals ('set 1' and 'set 2', respectively),

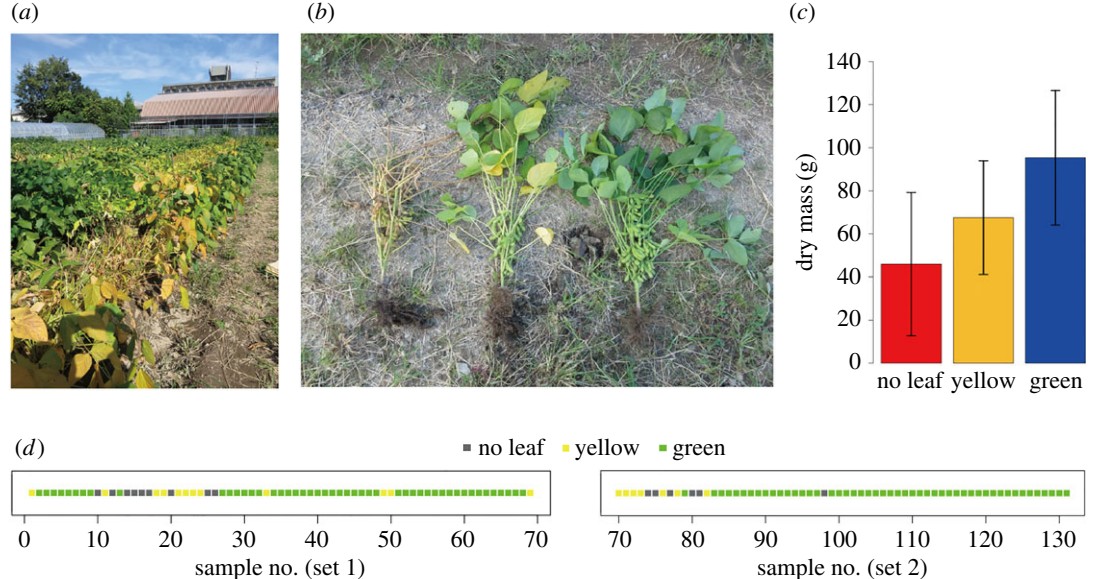

**Figure 1.** Study site and soybeans. (*a*) Soybean field in which sampling was conducted. (*b*) Soybean states. Soybean individuals were classified into three categories: those heavily attacked by root-knot nematodes ('no leaf'; left), those exhibited normal growth ('green'; right) and those showing intermediate characters ('yellow'; middle). (*c*) Relationship between soybean states and biomass. Dry mass significantly differed among 'no leaf', 'yellow' and 'green' soybean individuals (ANOVA; $F_2 = 20.5$, $p < 00001$). (*d*) Spatial distribution of 'no leaf', 'yellow' and 'green' soybean individuals. Sampling sets 1 and 2 are shown separately.

respectively, were sampled at every other position (i.e. 30 cm intervals; figure 1) on 7 October 2016. The sampled soybean individuals were classified into three categories: normal individuals with green leaves (green), individuals with yellow leaves (yellow) and those with no leaves (no leaf) (figure 1*a*–*c*). Among them, 'green' individuals exhibited normal growth, while 'no leaf' individuals were heavily infected by root-knot nematodes: 'yellow' individuals showed intermediate characters. In total, 97 'green', 19 'yellow' and 15 'no leaf' individuals were sampled (figure 1*d*).

For each individual, two segments of 5-cm terminal roots and rhizosphere soil were collected from around 10 cm below the soil surface. The root and soil samples were transferred into a cool box in the field and then stored at −80°C until DNA extraction in the laboratory. The above-ground bodies of the individuals were placed in drying ovens at 80°C for 72 h to measure dry mass. The dry mass data indicated that 'green', 'yellow' and 'no leaf' soybean individuals differed significantly in their biomass (figure 1*c*).

## 2.2. DNA extraction, PCR and sequencing

The root segments of each individual were transferred to a 15 ml tube and washed in 70% ethanol by vortexing for 10 s. The samples were then transferred to a new 15 ml tube and then washed again in 70% ethanol by sonication (42 Hz) for 5 min. After an additional sonication wash in a new tube, one of the two root segments was dried and placed in a 1.2 ml tube for each soybean individual. DNA extraction was then performed with a cetyltrimethylammonium bromide method [23] after pulverizing the roots with 4 mm zirconium balls at 25 Hz for 3 min using a TissueLyser II (Qiagen).

For DNA extraction from the rhizosphere soil, the ISOIL for Beads Beating kit (Nippon Gene) was used as instructed by the manufacturer. For each sample, 0.5 g of soil was placed into a 2 ml microtube of the ISOIL kit. To increase the yield of DNA, 10 mg of skim milk powder (Wako, 198–10605) was added to each sample [24].

For each of the root and soil samples, the 16S rRNA V4 region of the prokaryotes and the internal transcribed spacer 1 (ITS1) region of fungi were amplified. The PCR of the 16S rRNA region was performed with the forward primer 515f [25] fused with 3–6-mer Ns for improved Illumina sequencing quality [26] and the forward Illumina sequencing primer (5'- TCG TCG GCA GCG TCA GAT GTG TAT AAG AGA CAG- (3–6-mer Ns) – (515f) -3') and the reverse primer 806rB [27] fused with 3–6-mer Ns and the reverse sequencing primer (5'- GTC TCG TGG GCT CGG AGA TGT GTA

TAA GAG ACA G (3–6-mer Ns) - (806rB) -3′) (0.2 µM each). To prevent the amplification of mitochondrial and chloroplast 16S rRNA sequences, specific peptide nucleic acids (mPNA and pPNA; [26]) (0.25 µM each) were added to the reaction mix of KOD FX Neo (Toyobo). The temperature profile of the PCR was 94°C for 2 min, followed by 35 cycles at 98°C for 10 s, 78°C for 10 s, 50°C for 30 s, 68°C for 50 s and a final extension at 68°C for 5 min. To prevent generation of chimaeric sequences, the ramp rate through the thermal cycles was set to $1°C \, s^{-1}$ [28]. Illumina sequencing adaptors were then added to respective samples in the supplemental PCR using the forward fusion primers consisting of the P5 Illumina adaptor, 8-mer indexes for sample identification [29] and a partial sequence of the sequencing primer (5′- AAT GAT ACG GCG ACC ACC GAG ATC TAC AC - (8-mer index) - TCG TCG GCA GCG TC -3′) and the reverse fusion primers consisting of the P7 adaptor, 8-mer indexes and a partial sequence of the sequencing primer (5′- CAA GCA GAA GAC GGC ATA CGA GAT - (8-mer index) - GTC TCG TGG GCT CGG -3′). KOD FX Neo was used with a temperature profile of 94°C for 2 min, followed by eight cycles at 98°C for 10 s, 55°C for 30 s, 68°C for 50 s (ramp rate $= 1°C \, s^{-1}$) and a final extension at 68°C for 5 min. The PCR amplicons of the 131 soybean individuals were then pooled after a purification/equalization process with the AMPureXP Kit (Beckman Coulter). Primer dimers, which were shorter than 200 bp, were removed from the pooled library by supplemental purification with AMPureXP: the ratio of AMPureXP reagent to the pooled library was set to 0.6 (v/v) in this process.

The PCR of fungal ITS1 region was performed with the forward primer ITS1F_KYO1 [30] fused with 3–6-mer Ns for improved Illumina sequencing quality [26] and the forward Illumina sequencing primer (5′- TCG TCG GCA GCG TCA GAT GTG TAT AAG AGA CAG- (3–6-mer Ns) – (ITS1F_KYO1) -3′) and the reverse primer ITS2_KYO2 [30] fused with 3–6-mer Ns and the reverse sequencing primer (5′- GTC TCG TGG GCT CGG AGA TGT GTA TAA GAG ACA G (3–6-mer Ns) - (ITS2_KYO2) -3′). The buffer and polymerase system of KOD FX Neo was used with a temperature profile of 94°C for 2 min, followed by 35 cycles at 98°C for 10 s, 50°C for 30 s, 68°C for 50 s, and a final extension at 68°C for 5 min. Illumina sequencing adaptors and 8-mer index sequences were then added in the second PCR as described above. The amplicons were purified and pooled as described above.

The sequencing libraries of the prokaryote 16S and fungal ITS regions were processed in an Illumina MiSeq sequencer (run center: KYOTO-HE; 15% PhiX spike-in). Because the quality of forward sequences is generally higher than that of reverse sequences in Illumina sequencing, we optimized the MiSeq run setting in order to use only forward sequences. Specifically, the run length was set at 271 forward (R1) and 31 reverse (R4) cycles in order to enhance forward sequencing data: the reverse sequences were used only for discriminating between 16S and ITS1 sequences based on the sequences of primer positions.

## 2.3. Bioinformatics

The raw sequencing data were converted into FASTQ files using the program bcl2fastq 1.8.4 distributed by Illumina. The output FASTQ files were demultiplexed with the program Claident v0.2.2017.05.22 [31,32], by which sequencing reads whose 8-mer index positions included nucleotides with low (less than 30) quality scores were removed. The sequencing data were deposited to DNA Data Bank of Japan (DDBJ) (DRA006845). Only forward sequences were used in the following analyses after removing low-quality 3′-ends using Claident. Noisy reads [31] were subsequently discarded and then denoised dataset consisting of 2 041 573 16S and 1 325 199 ITS1 reads were obtained.

For each dataset of 16S and ITS1 regions, filtered reads were clustered with a cut-off sequencing similarity of 97% using the program VSEARCH [33] as implemented in Claident. The operational taxonomic units (OTUs) representing less than 10 sequencing reads were subsequently discarded. The molecular identification of the remaining OTUs was performed based on the combination of the query-centric auto-$k$-nearest neighbour (QCauto) method [32] and the lowest common ancestor (LCA) algorithm [34] as implemented in Claident. Note that taxonomic identification results based on the combination of the QCauto search and the LCA taxonomic assignment are comparable to, or sometimes more accurate than, those with the alternative approaches [32,35,36]. In total, 5351 prokatyote (bacterial or archaeal) OTUs and 1039 fungal OTUs were obtained for the 16S and ITS1 regions, respectively (electronic supplementary material, data S1). The UNIX codes used in the above bioinformatic pipeline are available as electronic supplementary material, data S2.

For each combination of target region (16S or ITS1) and sample type (root or soil), we obtained a sample × OTU matrix, in which a cell entry depicted the number of sequencing reads of an OTU in a sample (electronic supplementary material, data S3). The cell entries whose read counts represented less than 0.1% of the total read count of each sample were removed to minimize effects of

PCR/sequencing errors [37]. The filtered matrix was then rarefied to 1000 reads per sample using the 'rrarefy' function of the vegan 2.4–1 package [38] of R 3.4.3 [39]. Samples with less than 1000 reads were discarded in this process: the numbers of samples in the rarefied sample × OTU matrices were 119, 128, 117 and 128 for root prokaryote, root fungal, soil prokaryote and soil fungal matrices, respectively (electronic supplementary material, data S4).

## 2.4. Prokaryote and fungal community structure

Relationship between the number of sequencing reads and that of detected OTUs was examined for each dataset (root prokaryote, root fungal, soil prokaryote or soil fungal dataset) with the 'rarecurve' function of the R vegan package. Likewise, relationship between the number of samples and that of OTUs was examined with the vegan 'specaccum' function. For each dataset, difference in OTU compositions among 'green', 'yellow' and 'no leaf' soybean individuals was examined by the permutational analysis of variance (PERMANOVA; [40]) with the vegan 'adonis' function (10 000 permutations). To control effects of sampling positions (lines) on the community structure, the information of sampling sets (set 1 or set 2) was included as an explanatory variable in the PERMANOVA. The variation in OTU compositions was visualized with non-metric multidimensional scaling (NMDS) using the vegan 'metaMDS' function. To examine the potential relationship between root/soil microbial community structure and plant biomass, an additional PERMANOVA was performed for each dataset. The information of sampling sets was included in the models. To explore signs of spatial autocorrelation in the community data, a Mantel's correlogram analysis was performed with the vegan 'mantel.correlog' function. The 'Bray–Curtis' metric of β-diversity was used in the PERMANOVA, NMDS and Mantel's correlogram analyses.

## 2.5. Screening of host-state-specific OTUs

To explore prokaryote/fungal OTUs that preferentially occurred on/around 'green', 'yellow' or 'no leaf' soybean individuals, a randomization test was performed by shuffling the plant state labels in each of the root prokaryote, root fungal, soil prokaryote and soil fungal data matrices (100 000 permutations). We then evaluated preference of a prokaryote/fungal OTU ($i$) for a plant state ($j$) (green', 'yellow' or 'no leaf) as follows:

$$\text{Preference}(i, j) = \frac{[N_{\text{observed}}(i, j) - \text{Mean}(N_{\text{randomized}}(i, j))]}{\text{s.d.}(N_{\text{randomized}}(i, j))},$$

where $N_{\text{observed}}(i, j)$ denoted the mean number of the sequencing reads of OTU $i$ among state $j$ soybean samples in the original data, and the Mean ($N_{\text{randomized}}(i, j)$) and s.d. ($N_{\text{randomized}}(i, j)$) were the mean and standard deviation of the number of sequencing reads for the focal OTU–plant state combination across randomized matrices. Regarding this standardized preference index, values larger than three generally represent strong preferences (false discovery rate (FDR) < 0.05; see results of a previous study [35]): hence, we listed OTUs whose preference values exceeded three.

## 2.6. Microbe – microbe networks

To examine how prokaryote and fungal OTUs co-occurred in root or soil samples, a co-abundance network analysis was performed based on the sparse inverse covariance estimation for ecological association inference (Spiec-Easi) method [41]. In each of the root and soil sample analyses, the input data matrix was prepared by merging the sample × OTU matrices of prokaryotes and fungi. As inferences of co-abundance patterns were unavailable for rare OTUs, only the OTUs detected from 30 or more samples were retained in the input matrices. For each of the root and soil data matrices, a co-abundance analysis was performed with the 'spiec.easi' function of the R 'SpiecEasi' package [41]. The networks depicting the co-abundance patterns were drawn using the R 'igraph' package [42].

# 3. Results

## 3.1. Prokaryotes and fungal community structure

On average, 107.9 (s.d. = 18.0), 25.4 (s.d. = 8.9), 172.5 (s.d. = 17.3) and 78.3 (s.d. = 10.5) OTUs per sample were observed, respectively, from the root prokaryote, root fungal, soil prokaryote and soil fungal dataset

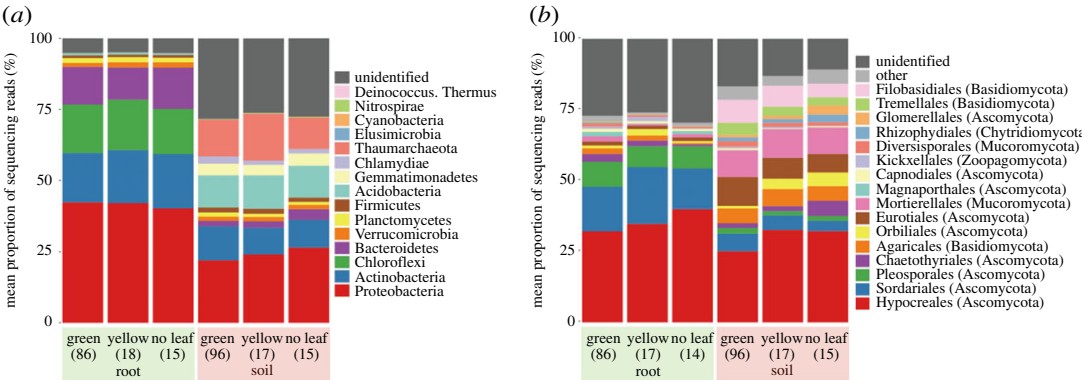

**Figure 2.** Prokaryote and fungal community structure. (*a*) Phylum-level compositions of prokaryotes in the root and soil datasets. Mean proportions of sequencing reads are shown for respective taxa. The numbers of the samples from which sequencing data were successfully obtained are shown in the parentheses. (*b*) Order-level compositions of fungi in the root and soil datasets.

after filtering and rarefaction steps (electronic supplementary material, figure S2). The total number of OTUs observed was 1387, 346, 1191 and 769 for the root prokaryote, root fungal, soil prokaryote and soil fungal datasets, respectively (electronic supplementary material, figure S3).

In the soybean field, the prokaryote community on roots was dominated by the bacterial classes Proteobacteria, Actinobacteria, Chloroflexi and Bacteroidetes, while that of rhizosphere soil consisted mainly of Proteobacteria, Actinobacteria and Acidobacteria, and the archaeal lineage Thaumarchaeota (figure 2*a*). The fungal community of roots was dominated by the fungal orders Hypocreales, Sordariales, Plesporales, while that of soil consisted mainly of Hypocreales, Agaricales, Eurotiales, Mortierellales and Filobasidiales (figure 2*b*). Regarding the order-level compositions of fungi in the rhizosphere soil, the proportion of Orbiliales reads was much higher in 'yellow' (3.62%) and 'no leaf' (4.82%) soybean individuals than in 'green' ones (0.89%) (figure 2). The genus level compositions of the samples are shown in electronic supplementary material, figure S4.

In each dataset (i.e. root prokaryote, root fungal, soil prokaryote or soil fungal data), microbial community structure varied among 'green', 'yellow' or 'no leaf' soybean individuals, although the effects of sampling sets on the community structure were much stronger (figure 3). Even within each sampling set, spatial autocorrelations of bacterial/fungal community structure were observed (electronic supplementary material, figure S5). Significant relationships between microbial community structure and soybean biomass were observed in the soil prokaryote and soil fungal datasets but not in the root prokaryote and root fungal datasets (table 1).

## 3.2. Screening of host-state-specific OTUs

In the root microbiome, only an unidentified fungal OTU showed a strong preference for 'green' soybean individuals, while 18 bacterial and four fungal OTUs occurred preferentially on 'no leaf' host individuals (table 2; electronic supplementary material, figure S6). The list of the bacteria showing preferences for 'no leaf' soybean individuals included OTUs whose 16S rRNA sequences were allied to those of *Dyella*, *Herbaspirillum*, *Labrys*, *Phenylobacterium*, *Gemmata*, *Chitinophaga*, *Pedobacter*, *Niastella* and *Streptomyces* (table 2). The four fungal OTUs showing preferences for 'no leaf' hosts were unidentified basidiomycetes (table 2).

In the rhizosphere soil microbiome, seven prokaryote OTUs, including those belonging to Chloroflexi (e.g. *Sphaerobacteraceae* sp.) and Proteobacteria (*Kofleriaceae* sp.), occurred preferentially on 'green' host individuals (table 3). Likewise, five fungal OTUs, including those allied to basidiomycete yeasts in the genera *Solicoccozyma* and *Saitozyma*, showed preferences for 'green' soybean individuals (table 3). Results also revealed that 26 bacterial and 11 fungal OTUs had biased distributions in the rhizosphere of 'no leaf' soybean individuals (table 3). The list of microbes showing preferences for 'no leaf' hosts included OTUs allied to bacteria in the genera *Pesudomonas*, *Nevskia*, *Cellvibrio*, *Massilia*, *Duganella*, *Novosphingobium*, *Mucilaginibacter* and *Flavobacterium* and OTUs allied to fungi in the genera *Burgoa*, *Clonostachys*, *Plectosphaerella*, *Xylaria*, *Dactylellina*, *Talaromyces*, *Cladosporium*, *Alternaria* and *Peniophora* (table 3). The list of microbes that preferentially occurred on 'no leaf' hosts involved OTUs with high sequence similarity to the nematophagous fungi, *Clonostachys rosea* (Hypocreales) and *Dactylellina* sp.

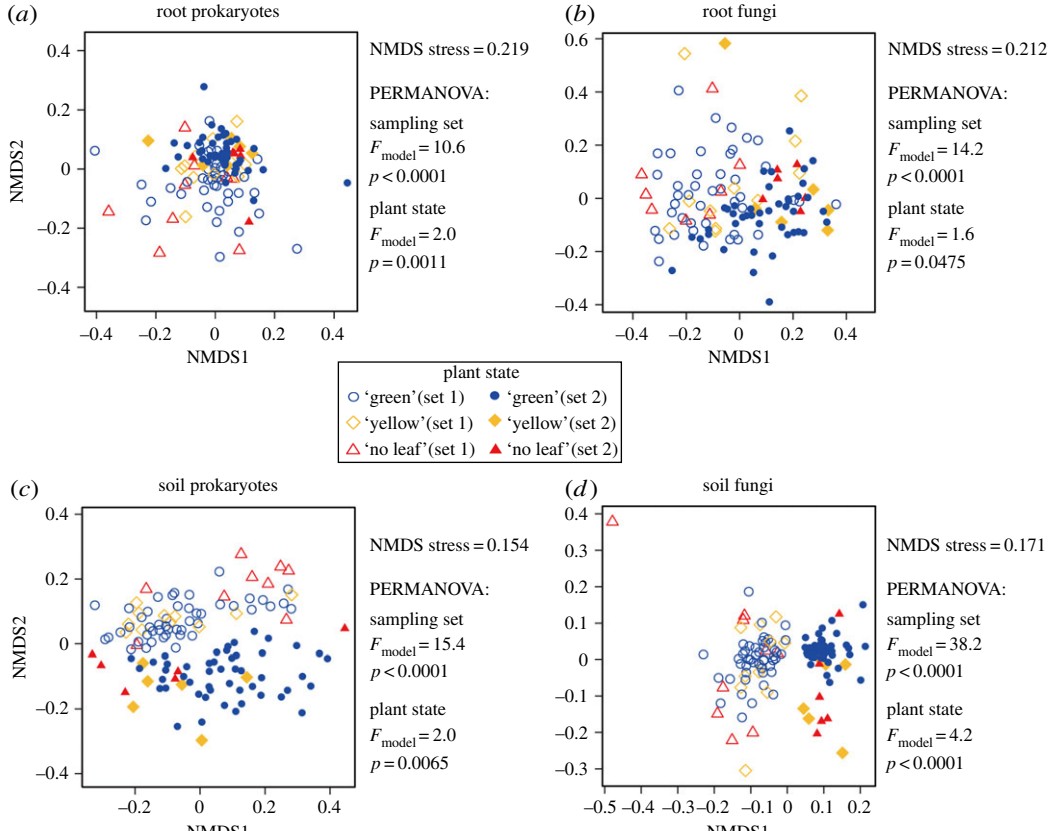

**Figure 3.** Diversity of microbiome structures among samples. (*a*) NMDS of the root prokaryote dataset. The results of the PERMANOVA, in which sampling set (set 1 or set 2) and plant state (green, yellow or no leaf) were included as explanatory variables, are shown. (*b*) NMDS of the root fungal dataset. (*c*) NMDS of the soil prokaryote dataset. (*d*) NMDS of the soil fungal dataset.

**Table 1.** Relationship between prokaryote/fungal community structure and the biomass of soybean individuals. For each dataset (i.e. root prokaryote, root fungal, soil prokaryote or soil fungal data), a PEMANOVA model of community structure was constructed. The information of the sampling set (set 1 or set 2) and the dry mass of host soybean individuals were included as explanatory variables.

| variable | d.f. | $F_{model}$ | $p$ |
|---|---|---|---|
| **root prokaryotes** | | | |
| sampling set | 1 | 10.4 | <0.0001 |
| dry mass | 1 | 1.3 | 0.1139 |
| **root fungi** | | | |
| sampling set | 1 | 14.0 | <0.0001 |
| dry mass | 1 | 0.6 | 0.8267 |
| **soil prokaryotes** | | | |
| sampling set | 1 | 15.4 | <0.0001 |
| dry mass | 1 | 3.1 | 0.002 |
| **soil fungi** | | | |
| sampling set | 1 | 36.7 | <0.0001 |
| dry mass | 1 | 2.2 | 0.0145 |

(Orbiliales) (table 3). The reads of the *Clonostachys* (F_0257) and *Dactylellina* (F_0163) OTUs, respectively, represented 9.5% and 3.5% of the sequencing reads of 'no leaf' samples (electronic supplementary material, data S5). The indices of preferences for 'yellow' soybean individuals are shown in electronic supplementary material, data S5.

**Table 2.** Prokaryote and fungal OTUs showing strong preferences for host states in the root microbiome datasets. The prokaryote/fungal OTUs that showed strong preferences for 'green' or 'no leaf' soybean individuals (preference value ≥ 3) are shown. The taxonomic assignment results based on the QCauto−LCA pipeline are shown with the top-hit results of NCBI BLAST searches. The OTU code starting with P (P_xxxx) and F (F_xxxx) are prokaryotes and fungi, respectively.

| OTU | phylum | class | order | family | genus | NCBI top hit | accession | cover (%) | Identity () |
|---|---|---|---|---|---|---|---|---|---|
| green | | | | | | | | | |
| F_0437 | Ascomycota | — | — | — | — | *Knufia* sp. | KP235641.1 | 83 | 98 |
| no leaf | | | | | | | | | |
| P_3453 | Proteobacteria | Gammaproteobacteria | Xanthomonadales | Rhodanobacteraceae | — | *Dyella marensis* | LN890104.1 | 100 | 99 |
| P_3207 | Proteobacteria | Gammaproteobacteria | Legionellales | Coxiellaceae | *Aquicella* | *Aquicella siphonis* | NR_025764.1 | 100 | 94 |
| P_2827 | Proteobacteria | Betaproteobacteria | — | — | — | *Duganella zoogloeoides* | KT983992.1 | 100 | 100 |
| P_2733 | Proteobacteria | Betaproteobacteria | Burkholderiales | Oxalobacteraceae | *Herbaspirillum* | *Herbaspirillum chlorophenolicum* | MG571754.1 | 100 | 100 |
| P_2590 | Proteobacteria | Alphaproteobacteria | — | — | — | *Croceicoccus mobilis* | NR_152701.1 | 100 | 88 |
| P_2481 | Proteobacteria | Alphaproteobacteria | Rickettsiales | Rickettsiaceae | — | *Rickettsia japonica* | KU586263.1 | 100 | 91 |
| P_2279 | Proteobacteria | Alphaproteobacteria | Rhizobiales | Xanthobacteraceae | *Labrys* | *Labrys monachus* | KT694157.1 | 100 | 100 |
| P_2042 | Proteobacteria | Alphaproteobacteria | Caulobacterales | Caulobacteraceae | *Phenylobacterium* | *Phenylobacterium* sp. | JX458410.1 | 100 | 99 |
| P_3664 | Proteobacteria | — | — | — | — | *Desulfofrigus oceanense* | AB568590.1 | 97 | 93 |
| P_3658 | Proteobacteria | — | — | — | — | *Rudaea* sp. | KM253197.1 | 100 | 85 |
| P_1748 | Planctomycetes | Planctomycetia | Planctomycetales | Gemmataceae | *Gemmata* | *Gemmata* sp. | GQ889445.1 | 100 | 99 |
| P_1278 | Chloroflexi | Thermomicrobia | — | — | — | *Sphaerobacter thermophilus* | AJ871227.1 | 100 | 92 |
| P_1058 | Bacteroidetes | — | — | — | — | *Chitinophaga polysaccharea* | MG322237.1 | 100 | 92 |
| P_1049 | Bacteroidetes | — | — | — | — | *Pedobacter terrae* | MG819444.1 | 100 | 98 |
| P_0994 | Bacteroidetes | — | — | — | — | *Chitinophaga terrae* | LN890054.1 | 100 | 95 |

(*Continued.*)

**Table 2.** (*Continued.*)

| OTU | phylum | class | order | family | genus | NCBI top hit | accession | cover (%) | Identity () |
|-----|--------|-------|-------|--------|-------|--------------|-----------|-----------|-------------|
| P_0887 | Bacteroidetes | Chitinophagia | Chitinophagales | Chitinophagaceae | *Niastella* | *Niastella koreensis* | NR_074595.1 | 100 | 100 |
| P_0498 | Actinobacteria | Actinobacteria | Streptomycetales | Streptomycetaceae | — | *Streptomyces albiaxialis* | KP170480.1 | 100 | 98 |
| P_0444 | Actinobacteria | Actinobacteria | Streptomycetales | Streptomycetaceae | *Streptomyces* | *Streptomyces olivaceoviridis* | KP823723.1 | 100 | 98 |
| F_0796 | Basidiomycota | — | — | — | — | *Classiculaceae* sp. | KY548838.1 | 92 | 84 |
| F_0792 | Basidiomycota | — | — | — | — | *Classiculaceae* sp. | KY548838.1 | 92 | 83 |
| F_0790 | Basidiomycota | — | — | — | — | *Classiculaceae* sp. | KY548838.1 | 91 | 83 |
| F_0786 | Basidiomycota | — | — | — | — | *Classiculaceae* sp. | KY548838.1 | 90 | 84 |

**Table 3.** Prokaryote and fungal OTUs showing strong preferences for host states in the soil microbiome datasets. The prokaryote/fungal OTUs that that showed strong preferences for 'green' or 'no leaf' soybean individuals (preference value ≥ 3) are shown. The taxonomic assignment results based on the QCauto−LCA pipeline are shown with the top-hit results of NCBI BLAST searches. The OTU code starting with P (P_xxxx) and F (F_xxxx) are prokaryotes and fungi, respectively.

| OTU | phylum | class | order | family | genus | NCBI top hit | accession | cover (%) | Identity (%) |
|---|---|---|---|---|---|---|---|---|---|
| green | | | | | | | | | |
| P_0697 | Actinobacteria | — | — | — | — | *Gaiella occulta* | NR_118138.1 | 100 | 91 |
| P_1264 | Chloroflexi | Thermomicrobia | Sphaerobacterales | Sphaerobacteraceae | *Sphaerobacter* | *Shewanella fodinae* | FM887036.1 | 98 | 84 |
| P_1281 | Chloroflexi | Thermomicrobia | — | — | — | *Thermomicrobium carboxidum* | NR_134218.1 | 100 | 87 |
| P_2949 | Proteobacteria | Deltaproteobacteria | Myxococcales | Kofleriaceae | *Haliangium* | *Kofleria flava* | HF937255.1 | 100 | 91 |
| P_3762 | — | — | — | — | — | *Planctomycetales bacterium* | AY673390.1 | 98 | 94 |
| P_3715 | — | — | — | — | — | *Brochothrix thermosphacta* | MG807446.1 | 99 | 86 |
| P_0032 | — | — | — | — | — | *Nitrosocosmicus exaquare* | CP017922.1 | 100 | 99 |
| F_0477 | Ascomycota | — | — | — | — | No significant match | — | — | — |
| F_0141 | Ascomycota | Eurotiomycetes | — | — | — | *Penicillium clavigerum* | NR_121317.1 | 100 | 81 |
| F_0700 | Basidiomycota | Tremellomycetes | Filobasidiales | Piskurozymaceae | *Solicoccozyma* | *Solicoccozyma terreus* | KY102958.1 | 100 | 100 |
| F_0734 | Basidiomycota | Tremellomycetes | Tremellales | Trimorphomycetaceae | *Saitozyma* | *Saitozyma podzolica* | KY102943.1 | 82 | 99 |
| F_0738 | Basidiomycota | Tremellomycetes | Tremellales | Trimorphomycetaceae | *Saitozyma* | *Saitozyma podzolica* | KY102943.1 | 84 | 99 |

(*Continued.*)

**Table 3.** (*Continued.*)

| OTU | phylum | class | order | family | genus | NCBI top hit | accession | cover (%) | Identity (%) |
|---|---|---|---|---|---|---|---|---|---|
| no leaf | | | | | | | | | |
| P_3294 | Proteobacteria | Gammaproteobacteria | Pseudomonadales | Pseudomonadaceae | Pseudomonas | Pseudomonas psychrotolerans | KY623077.1 | 100 | 100 |
| P_3256 | Proteobacteria | Gammaproteobacteria | Nevskiales | Sinobacteraceae | Nevskia | Nevskia persephonica | JQ710442.1 | 97 | 99 |
| P_3189 | Proteobacteria | Gammaproteobacteria | Cellvibrionales | Cellvibrionaceae | Cellvibrio | Cellvibrio mixtus | KC329916.1 | 100 | 100 |
| P_3308 | Proteobacteria | Gammaproteobacteria | — | — | — | Steroidobacter sp. | KP185148.1 | 100 | 95 |
| P_3093 | Proteobacteria | Deltaproteobacteria | Myxococcales | — | — | Sorangiineae bacterium | JF719608.1 | 100 | 94 |
| P_3004 | Proteobacteria | Deltaproteobacteria | Myxococcales | Polyangiaceae | Byssovorax | Polyangium spumosum | KX572839.2 | 100 | 97 |
| P_3114 | Proteobacteria | Deltaproteobacteria | — | — | — | Stigmatella hybrida | KX572784.2 | 100 | 91 |
| P_2747 | Proteobacteria | Betaproteobacteria | Burkholderiales | Oxalobacteraceae | — | Massilia kyonggiensis | NR_126273.1 | 100 | 100 |
| P_2827 | Proteobacteria | Betaproteobacteria | — | — | — | Duganella radicis | LC191531.1 | 100 | 100 |
| P_2552 | Proteobacteria | Alphaproteobacteria | Sphingomonadales | Sphingomonadaceae | — | Novosphingobium sediminicola | KX987160.1 | 100 | 100 |
| P_1637 | Gemmatimonadetes | Gemmatimonadetes | Gemmatimonadales | Gemmatimonadaceae | Gemmatimonas | Gemmatimonas aurantiaca | KF228166.1 | 100 | 93 |
| P_1544 | Gemmatimonadetes | Gemmatimonadetes | Gemmatimonadales | Gemmatimonadaceae | Gemmatimonas | Gemmatimonas sp. | LN876485.1 | 100 | 89 |
| P_0962 | Bacteroidetes | Sphingobacteriia | Sphingobacteriales | Sphingobacteriaceae | Mucilaginibacter | Mucilaginibacter gotjawali | AP017313.1 | 100 | 99 |
| P_0892 | Bacteroidetes | Chitinophagia | Chitinophagales | Chitinophagaceae | — | Ferruginibacter profundus | NR_148259.1 | 100 | 88 |

(*Continued.*)

**Table 3.** (*Continued.*)

| OTU | phylum | class | order | family | genus | NCBI top hit | accession | cover (%) | Identity (%) |
|---|---|---|---|---|---|---|---|---|---|
| P_1095 | Bacteroidetes | — | — | — | — | *Flavisolibacter ginsengisoli* | NR_041500.1 | 100 | 95 |
| P_1051 | Bacteroidetes | — | — | — | — | *Flavobacterium lindanitolerans* | KP875419.1 | 100 | 100 |
| P_1008 | Bacteroidetes | — | — | — | — | *Solitalea canadensis* | CP003349.1 | 100 | 88 |
| P_0652 | Actinobacteria | Thermoleophilia | Solirubrobacterales | Solirubrobacteraceae | *Solirubrobacter* | *Solirubrobacter phytolaccae* | NR_133858.1 | 99 | 92 |
| P_5169 | — | — | — | — | — | *Desulfotomaculum nigrificans* | NR_074579.1 | 97 | 85 |
| P_5087 | — | — | — | — | — | *Stenotrophobacter roseus* | NR_146022.1 | 99 | 97 |
| P_4649 | — | — | — | — | — | *Alkalilimnicola ehrlichii* | NR_074775.1 | 99 | 81 |
| P_4607 | — | — | — | — | — | *Verrucomicrobia* | JF488114.1 | 100 | 92 |
| P_4606 | — | — | — | — | — | *Ruminococcus flavefaciens* | KX155563.1 | 99 | 83 |
| P_4595 | — | — | — | — | — | *Moorella thermoacetica* | NR_043076.1 | 97 | 84 |
| P_3783 | — | — | — | — | — | *Fimbriimonas ginsengisoli* | CP007139.1 | 100 | 88 |
| P_3739 | — | — | — | — | — | *Solibacter usitatus* | GQ287461.1 | 100 | 88 |
| F_0866 | Mucoromycota | Glomeromycetes | — | — | — | *Acaulospora delicata* | JF439203.1 | 45 | 95 |
| F_0620 | Basidiomycota | Agaricomycetes | Polyporales | — | *Burgoa* | *Burgoa anomala* | AB972783.1 | 100 | 100 |

**13**

**Table 3.** (*Continued.*)

| OTU | phylum | class | order | family | genus | NCBI top hit | accession | cover (%) | Identity (%) |
|---|---|---|---|---|---|---|---|---|---|
| F_0785 | Basidiomycota | — | — | — | — | *Radulomyces copelandii* | MG722738.1 | 87 | 99 |
| F_0257 | Ascomycota | Sordariomycetes | Hypocreales | Bionectriaceae | *Clonostachys* | *Clonostachys rosea* | KY320599.1 | 100 | 100 |
| F_0237 | Ascomycota | Sordariomycetes | Glomerellales | Plectosphaerellaceae | — | *Plectosphaerella plurivora* | KU204617.1 | 98 | 99 |
| F_0413 | Ascomycota | Sordariomycetes | — | — | — | *Xylariales* sp. | KY031690.1 | 100 | 100 |
| F_0163 | Ascomycota | Orbiliomycetes | Orbiliales | Orbiliaceae | *Dactylellina* | *Dactylellina* aff. *ellipsospora* | KT215204.1 | 100 | 99 |
| F_0131 | Ascomycota | Eurotiomycetes | Eurotiales | — | — | *Talaromyces verruculosus* | KC937053.1 | 100 | 98 |
| F_0003 | Ascomycota | Dothideomycetes | Capnodiales | Cladosporiaceae | *Cladosporium* | *Cladosporium cladosporioides* | MG946764.1 | 100 | 100 |
| F_0482 | Ascomycota | — | — | — | — | *Alternaria alternata* | KY367499.2 | 100 | 100 |
| F_0973 | — | — | — | — | — | *Peniophora incarnata* | EU918698.1 | 100 | 98 |

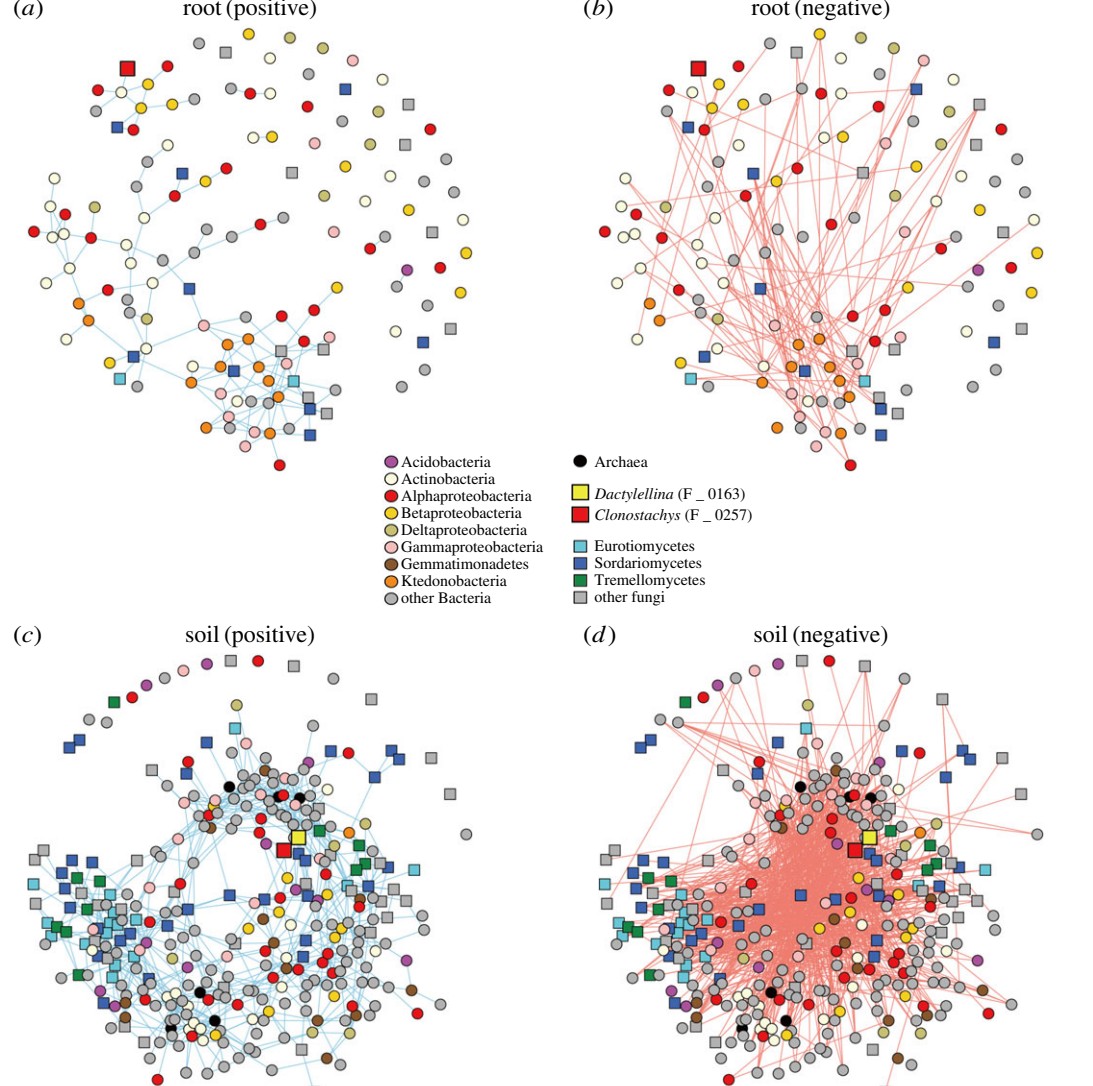

**Figure 4.** Microbe–microbe co-abundance networks. (*a*) Positive co-abundance network of the root microbiome data. Pairs of OTUs linked by a blue line frequently co-occurred in the same soybean samples. (*b*) Negative co-abundance network of the root microbiome data. Pairs of OTUs linked by a red line rarely co-occurred in the same soybean samples. (*c*) Positive co-abundance network of the soil microbiome data. (*d*) Negative co-abundance network of the soil microbiome data.

## 3.3. Microbe–microbe networks

The structures of microbe–microbe networks (figure 4) were more complicated in the soil microbiome data (figure 4*c*,*d*) than in the root microbiome data (figure 4*a*,*b*). Within the network representing co-abundance of microbes across root samples, the *Clonostachys* OTU (F_0257) had a significant link with a *Streptomyces* OTU, while *Dactylellina* was absent from the root microbiome network data (figure 4*a*). Within the positive co-abundance network of the rhizosphere soil microbiome (figure 4*c*), the *Clonostachys* (F_0257) and *Dactylellina* (F_0163) nematophagous fungal OTUs were connected with each other (table 4). In addition, the *Clonostachys* OTU was linked with two bacterial OTUs (*Ralstonia* and Rhizobiales) and fungal OTUs in the genera *Calonectria* and *Purpureocillium* (table 4). Likewise, the *Dactylellina* OTU was connected also with two Alphaproteobacterial OTUs and a bacterial OTU allied to *Nitrospira japonica* as well as fungal OTUs in the genera *Rhizophydium*, *Pochonia* and *Purpureocillium* (table 4).

## 4. Discussion

Based on Illumina sequencing, we compared root-associated/rhizosphere microbial communities between soybean individuals infected by root-knot nematodes and those showing no symptoms. The

**Table 4.** Prokaryote/fungal OTUs linked to nematophagous fungi in the microbe–microbe co-abundance networks (figure 4a,c), the prokaryote/fungal OTUs that showed positive co-abundance patterns with *Clonostachys* (F_0257) and *Dactylellina* (F_0163) nematophagous fungal OTUs are listed. The taxonomic assignment results based on the QCauto–LCA pipeline are shown with the top-hit results of NCBI BLAST searches. The OTU code starting with P (P_xxxx) and F (F_xxxx) are prokaryotes and fungi, respectively.

| OTU | phylum | class | order | family | genus | NCBI top hit | accession | Cover (%) | Identity (%) |
|---|---|---|---|---|---|---|---|---|---|
| root: OTUs linked to *Clonostachys rosea* (F_0257) | | | | | | | | | |
| P_0510 | Actinobacteria | Actinobacteria | Streptomycetales | Streptomycetaceae | — | *Streptomyces nigrogriseolus* | MG984076.1 | 100 | 98 |
| soil: OTUs linked to *Clonostachys rosea* (F_0257) | | | | | | | | | |
| P_2689 | Proteobacteria | Betaproteobacteria | Burkholderiales | Burkholderiaceae | *Ralstonia* | *Ralstonia pickettii* | MF179868.1 | 100 | 100 |
| P_2243 | Proteobacteria | Alphaproteobacteria | Rhizobiales | — | — | *Pedomicrobium americanum* | NR_104908.1 | 100 | 90 |
| F_0163 | Ascomycota | Orbiliomycetes | Orbiliales | Orbiliaceae | *Dactylellina* | *Dactylellina* aff. *ellipsospora* | KT215204.1 | 100 | 99 |
| F_0278 | Ascomycota | Sordariomycetes | Hypocreales | Nectriaceae | *Calonectria* | *Calonectria zuluensis* | NR_137728.1 | 97 | 100 |
| F_0310 | Ascomycota | Sordariomycetes | Hypocreales | Ophiocordycipitaceae | — | *Purpureocillium lilacinum* | KP691502.1 | 100 | 100 |
| soil: OTUs linked to *Dactylellina sp.* (F_0163) | | | | | | | | | |
| P_2443 | Proteobacteria | Alphaproteobacteria | Rhodospirillales | — | — | *Azospirillum brasilense* | KY010284.1 | 100 | 92 |
| P_2589 | Proteobacteria | Alphaproteobacteria | — | — | — | *Elstera litoralis* | KR856497.1 | 100 | 92 |
| P_3774 | — | — | — | — | — | *Nitrospira japonica* | LT828648.1 | 100 | 100 |
| F_0812 | Chytridiomycota | Chytridiomycetes | Rhizophydiales | Rhizophydiaceae | *Rhizophydium* | *Rhizophydium* sp. | AY349124.1 | 99 | 100 |
| F_0278 | Ascomycota | Sordariomycetes | Hypocreales | Nectriaceae | *Calonectria* | *Calonectria zuluensis* | NR_137728.1 | 97 | 100 |
| F_0265 | Ascomycota | Sordariomycetes | Hypocreales | Clavicipitaceae | *Pochonia* | *Pochonia chlamydosporia* | KY977543.1 | 100 | 100 |
| F_0257 | Ascomycota | Sordariomycetes | Hypocreales | Bionectriaceae | *Clonostachys* | *Clonostachys rosea* | KY320599.1 | 100 | 100 |
| F_0310 | Ascomycota | Sordariomycetes | Hypocreales | Ophiocordycipitaceae | — | *Purpureocillium lilacinum* | KP691502.1 | 100 | 100 |

results indicated that, in both soybean roots and rhizosphere soil, prokaryote and fungal community structures significantly varied depending on host plant states (figures 2 and 3). We further performed statistical analyses for screening prokaryote and fungal OTUs preferentially associated with infected and benign soybean host individuals (tables 2 and 3; figure 4). The results are based on purely descriptive data and hence they, in principle, are not direct evidences of interactions among plants, nematodes and microbiomes: i.e. causal relationship among those agents remains unknown. As this study provided only 'snap-shot' information of microbiome structure at the end of a growing season, we need to conduct further studies uncovering temporal microbiome dynamics throughout the growing season of soybeans. Nonetheless, as detailed below, the statistical analyses suggest assembly of diverse anti-nematode bacteria and fungi from indigenous microbial communities in the soybean field, providing a basis for exploring ways to reduce damage by root-knot nematodes with those indigenous functional microbes.

Within the root microbiome analysed, various taxonomic groups of bacteria preferentially occurred on 'no leaf' soybean samples (table 2). Among them, the genus *Streptomyces* is known to involve some species that suppress nematode populations, potentially used as biological control agents for root-knot nematodes [43–46]. By contrast, *Herbaspirillum*, *Rickettsia*, *Chitinophaga* and *Pedobacter* have been reported as symbionts of nematodes, potentially playing beneficial roles for host nematodes [47–49]. Results of these statistical analyses should be interpreted with caution, as they are likely to highlight not only prospective microbes potentially parasitizing on pests/pathogens, but also microbes that can form mutualistic interactions with disease agents.

Within the soybean rhizosphere soil microbiome, diverse taxonomic groups of not only bacteria, but also fungi preferentially occurred around 'no leaf' soybean individuals (table 3). Among them, *Pseudomonas* has been known to suppress root-knot nematode populations [50,51] potentially by producing hydrogen cyanide [52] or extracellular protease [53], but interactions with root-knot nematodes have not yet been examined for other bacteria preferentially found in the rhizosphere of 'no leaf' soybean individuals. Meanwhile, the list of the fungal OTUs frequently observed in the rhizosphere of 'no leaf' soybeans included some fungi whose ability to suppress nematode populations had been well documented (table 3). *Clonostachys rosea*, for example, has been known as a prospective biological control agent of plant- and animal-pathogenic nematodes [54,55]. An observational study based on green fluorescent protein imaging has indicated that the conidia of the fungus adhere to nematode cuticle and their germ tubes penetrate nematode bodies, eventually killing the invertebrate hosts [56]. The fungus is also known to produce a subtilisin-like extracellular protease, which plays an important role during the penetration of nematode cuticles [57]. Our analysis also highlighted a nematophagous fungus in the genus *Dactylellina* (teleomorph = *Orbilia*), which could capture juveniles of nematodes with hyphal traps [58]. Species in the genus and many other fungi in the order Orbiliales produce characteristic trap structures with their hyphae to prey on nematodes [59–61], often nominated as prospective biological control agents [62–64].

An additional analysis focusing on *Clonostachys* and *Dactylellina* highlighted bacteria and fungi that frequently co-occurred with the nematophagous fungi (figure 4). In the root microbiome, *Clonostachys* and a *Streptomyces* OTU showed positively correlated distributions across soybean samples (table 4). In the rhizosphere microbiome, *Clonostachys* and *Dactylellina* showed significant co-abundance patterns (table 4). Moreover, in the soil, the two nematophagous fungi co-occurred frequently with other taxonomic groups of nematophagous fungi such as *Purpureocillium*, *Pochonia* and *Rhizophydium* (table 4 and figure 5). Among them, fungi in the genus *Purpureocillium* (Hypocreales: Ophiocordycipitaceae) have been known to suppress plant parasitic nematodes, insect pests and oomycete phytopathogens [65–68]. Another Hypocreales genus, *Pochonia* (previously placed in the genus *Verticillium*; teleomorph = *Metacordyceps*; Clavicipitaceae) has been known as nematophagous as well and they can kill eggs and females of root-knot (*Meloidogyne* spp.) and cyst (*Globodera* spp.) nematodes [69–72]. Species in the chytrid genus *Rhizophydium* include species that use nematodes as parasites or saprophytes [73,74]. They are known to explore host nematodes in the form of zoospores [73]. All these results suggest that indigenous anti-nematode or nematophagous microbes can form consortia in soil ecosystems of soybean fields. It is important to note that the members of the consortia do not necessarily interact with each other directly: i.e. they may merely share habitat preferences [36,37,75]. However, the inferred structure of microbe–microbe networks helps us understand overall consequences of ecological processes in microbiomes [15].

Along with the consortia of anti-nematode microbes, an OTU in the genus *Calonectria*, which causes leaf blight, wilt and root rot of various plant species [76,77], was frequently observed (table 4). The phytopathogenic fungus might have attacked soybean individuals weakened by root-knot nematodes.

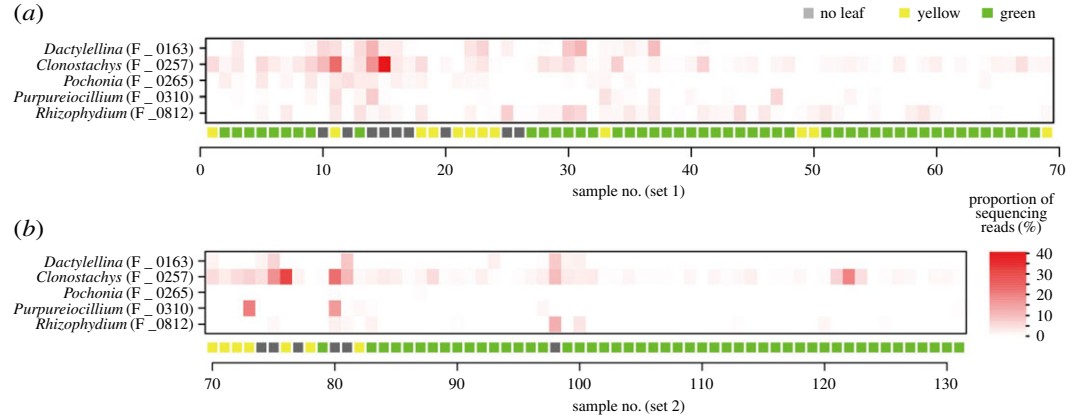

**Figure 5.** Spatial distribution of nematophagous fungal OTUs. (*a*) Sampling set 1. For each soybean individual, the proportions of sequencing reads representing nematophagous fungal OTUs are shown. (*b*) Sampling set 2.

Alternatively, *Calonectria* may have infected host soybeans earlier than root-knot nematodes, followed by the emergence of nematodes and their exploiters (i.e. anti-nematode microbes). Given that fungi can interact with each other both antagonistically and mutualistically in the soil [78,79], direct interactions between *Calonectria* and nematophagous fungi in the genera *Clonostachys*, *Dactylellina*, *Purpureocillium*, *Pochonia* and *Rhizophydium* are of particular interest. Studies examining potential interactions involving soybeans, root-knot nematodes, anti-nematode bacteria/fungi and *Calonectria* will help us understand ecological processes that structure consortia of nematophagous fungi.

Although this study did not evaluate potential effects of background environmental conditions (e.g. soil pH and inorganic nitrogen concentration) on microbiome structure, management of edaphic conditions are expected to have great impacts on dynamics of anti-nematode microbiomes. A number of studies have explored ways to suppress nematode populations by optimizing cropping systems [1]. Crop rotation, in which planting of a crop variety and that of nematode-resistant varieties/species are rotated, has been recognized as an effective technique for regulating root-knot and cyst nematode populations [8,80,81]. By contrast, long-term continual cropping in soybean monoculture fields can increase anti-nematode bacteria and fungi (e.g. *Pseudomonas*, *Purpureocillium* and *Pochonia*), potentially resulting in lowered densities of cyst nematodes [12]. Tillage regimes [9–11] and introduction of organic matter (e.g. alfalfa leaves or crop residue) [82–84] have great impacts on nematode densities in farmlands, but their effects vary considerably among studies [1]. In addition, because nematode-infected plant individuals can show highly aggregated distributions at a small spatial scale within a farmland (figure 1*d*), tillage can promote the spread of plant damaging nematodes [22]. Frequent tillage may have negative impacts on populations of nematophagous fungi as a consequence of hyphal fragmentation (cf. [85]), but such destructive effects on fungal communities have not yet been tested intensively. Given that microbiome structures were not taken into account in most previous studies evaluating effects of cropping systems on nematode suppression (but see [12,21]), more insights into the relationship between agroecosystem management and indigenous (native) microbiome dynamics are required for building reproducible ways to develop disease-suppressive soil.

We herein found that consortia of anti-nematode bacteria and fungi could develop at a small spatial scale within a field of soybeans infected by root-knot nematodes. Given the diversity of those anti-nematode microbes observed in this study, multiple biological control agents are potentially available *in situ* without introducing exogenous ones depending on base compositions and conditions of indigenous microbiomes. In this respect, design of cropping systems (e.g. crop rotations, tillage frequencies, and inputs of fertilizer or organic matter) is of particular importance in activating and maximizing ecosystem functions that stem from resident microbial diversity [15]. Because those indigenous microbes, in general, have adapted to local biotic and abiotic environments, their populations are expected to persist more stably than exogenous microbes artificially introduced to a target agroecosystem (see [19] for reviews of the success/failure of microbial introduction). Elucidating the relationship between cropping systems and microbiome processes is the key to designing disease-suppressive agroecosystems.

Ethics. The fieldwork and sampling of materials were permitted by Crop Science Laboratory, Graduate School of Agriculture, Kyoto University.

Data accessibility. Data are available from the electronic supplementary material, data S1–S5 and DNA DDBJ (DRA006845).

Authors' contributions. H.T. conceived and designed the work. H.T. and Y.T. performed fieldwork. H.T. conducted molecular experiment and analysed the data. H.T. wrote the manuscript with Y.T. All authors gave final approval for publication.

Competing interests. The authors declare that the research was conducted in the absence of any commercial or financial relationships that could be constructed as conflict of interest.

Funding. This work was financially supported by JSPS KAKENHI Grant no. (15KT0032) and JST PRESTO (JPMJPR16Q6) to H.T.

Acknowledgements. We thank Tatsuhiko Shiraiwa for his support in fieldwork and Mizuki Shinoda, Ko Mizushima, Sarasa Amma and Hiroki Kawai for their support in molecular experiments. We are also grateful to Ryoji Shinya for his advice on the biology of nematodes. We are also grateful to anonymous reviewers for their productive comments.

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
