## [Reviewer comments · Royal Society Open Science]

Review History

RSOS-181693.R0 (Original submission)

Review form: Reviewer 1

Is the manuscript scientifically sound in its present form?

Yes

Are the interpretations and conclusions justified by the results?

Yes

Is the language acceptable?

Yes

Is it clear how to access all supporting data?

Yes

Do you have any ethical concerns with this paper?

No

Have you any concerns about statistical analyses in this paper?

I do not feel qualified to assess the statistics

Recommendation?

Accept with minor revision (please list in comments)

Comments to the Author(s)

The study investigated bacterial and fungal communities in root and rhizosphere of healthy and diseased soybean plants, affected by root-knot nematodes, in two rows of a field. Several OTU were identified that preferentially occurred on healthy or affected plants, and their connections in microbe-microbe networks described. The experimental design was very good. The data are well presented and conclusions supported by the data. I suggest to better discuss the underlying mechanisms or consequences of preferential occurrence of OTU. Those OTU on healthy plants might indicate their role in protecting the plant from nematode attack. Preferential OTU on diseased plants might live on the nematodes (following nematode population dynamics but not controlling it), or simply profit from resource leakage of diseased roots.

Minor comments:

L. 31-37: add summary of OTU preferentially occurring on healthy plants; remove list of nemativororous species.

L. Is "awaited" the right word?

L. 132 square

L. 226, 228 randomized

L. 273 better describe this OTU: SH group in UNITE?

L. 348 plays

L. 364 lilacinum

L. 379 Calonectria

L. 534, 547, 549, 620 (

L. 695, 704 volume, pages missing

Review form: Reviewer 2**Is the manuscript scientifically sound in its present form?**

Yes

Are the interpretations and conclusions justified by the results?

Yes

Is the language acceptable?

Yes

Is it clear how to access all supporting data?

Yes

Do you have any ethical concerns with this paper?

No

Have you any concerns about statistical analyses in this paper?

I do not feel qualified to assess the statistics

Recommendation?

Major revision is needed (please make suggestions in comments)

Comments to the Author(s)

This paper dealt with relationship between microbial communities in rhizosphere and root of soybean plants and the infection by root-knot nematodes. Interestingly, authors sampling the soybean individual in the one field plot and separate the soybean individuals into three groups (normal, yellow and no leaf) corresponding to the infection of root-knot nematodes. Overall, the manuscript is well-written, methods and results are well presented and conclusions are fully justified.

However, the significantly weaknesses are that there were no any data on the nematode infections. It is quite easy to measure the root-knot index, which is quite necessary to explain how serious of the plants infected by nematode. Moreover, the infection of the nematodes (secondary-stage juveniles) occurred in the seedling stage of the plants. However, sampling was conducted at the late stage of the soybean. It is quite hard to correlate the microbes in the rhizosphere soil and roots to the nematode damaging. Although there are some relationships between some nematophagous microbes previously reported and nematode damaging, most of those microbes could survive as saprophytes in the soil. The other key issue is that root-knot nematode is kind of animals and many microbes can colonize on the different stages of root-knot nematode.

However, there was no any examination on the microbial community on nematode.

Following are some recent papers that may have reference value to help promote the significance of this manuscript:

The analysis of rhizosphere microbial communities during an invasion by *Ralstonia solanacearum* (Wei et al., *Ralstonia solanacearum* pathogen disrupts bacterial rhizosphere microbiome during an invasion, 2018), besides, the results on rhizosphere, root and cyst microbial communities in disease suppressive soil provide more insights into consortia of anti-nematode bacteria (Hussain et al., *Bacterial community assemblages in the rhizosphere soil, root endosphere and cyst of soybean cyst nematode-suppressive soil challenged with nematodes*, 2018).

Review form: Reviewer 3 (Paul Orwin)

Is the manuscript scientifically sound in its present form?

Yes

Are the interpretations and conclusions justified by the results?

Yes

Is the language acceptable?

No

Is it clear how to access all supporting data?

Yes

Do you have any ethical concerns with this paper?

No

Have you any concerns about statistical analyses in this paper?

I do not feel qualified to assess the statistics

Recommendation?

Accept with minor revision (please list in comments)

Comments to the Author(s)

I thought the overall science was sound, and the paper provides an interesting insight into the correlation between plant health, root-knot nematode colonization, and corresponding recruitment of nematophagous fungi and nematocidal bacteria to the root and rhizosphere. The paper makes a sound contribution to the literature on these complex interactions, and the authors are careful not to overinterpret their results. The only issue I found with the article was some minor formatting issues and some substantive grammatical concerns especially in the Summary, Introduction, and Discussion sections. These concerns are described in the attached review file (Appendix A).

Decision letter (RSOS-181693.R0)

09-Jan-2019

Dear Dr Toju,

The editors assigned to your paper ("Consortia of anti-nematode fungi and bacteria in the rhizosphere of soybean plants attacked by root-knot nematodes") have now received comments from reviewers. We would like you to revise your paper in accordance with the referee and Associate Editor suggestions which can be found below (not including confidential reports to the Editor). Please note this decision does not guarantee eventual acceptance.

Please submit a copy of your revised paper before 01-Feb-2019. Please note that the revision deadline will expire at 00.00am on this date. If we do not hear from you within this time then it will be assumed that the paper has been withdrawn. In exceptional circumstances, extensions may be possible if agreed with the Editorial Office in advance. We do not allow multiple rounds of revision so we urge you to make every effort to fully address all of the comments at this stage. If deemed necessary by the Editors, your manuscript will be sent back to one or more of the original reviewers for assessment. If the original reviewers are not available, we may invite new reviewers.

If your study uses humans or animals please include details of the ethical approval received, including the name of the committee that granted approval. For human studies please also detail

whether informed consent was obtained. For field studies on animals please include details of all permissions, licences and/or approvals granted to carry out the fieldwork.

- Data accessibility

If you wish to submit your supporting data or code to Dryad (<http://datadryad.org/>), or modify your current submission to dryad, please use the following link:
<http://datadryad.org/submit?journalID=RSOS&manu=RSOS-181693>

- Competing interests

- Authors' contributions

- Acknowledgements

- Funding statement

Kind regards,
Royal Society Open Science Editorial Office
Royal Society Open Science

on behalf of Dr Berat Haznedaroglu (Associate Editor) and Professor Kevin Padian (Subject Editor)

Editor's comments:

Please consider carefully all the comments of the reviewers, who are largely positive about the manuscript but do have some substantial issues that need to be addressed.

Additionally, please have a native speaker of English edit the manuscript; we will not be able to accept it with extensive grammatical errors. Thanks for your submission and best of luck with your revision.

Reviewers' Comments to Author:

Reviewer: 1

Comments to the Author(s)

The study investigated bacterial and fungal communities in root and rhizosphere of healthy and diseased soybean plants, affected by root-knot nematodes, in two rows of a field. Several OTU were identified that preferentially occurred on healthy or affected plants, and their connections in microbe-microbe networks described. The experimental design was very good. The data are well presented and conclusions supported by the data. I suggest to better discuss the underlying mechanisms or consequences of preferential occurrence of OTU. Those OTU on healthy plants might indicate their role in protecting the plant from nematode attack. Preferential OTU on diseased plants might live on the nematodes (following nematode population dynamics but not controlling it), or simply profit from resource leakage of diseased roots.

Minor comments:

L. 31-37: add summary of OTU preferentially occurring on healthy plants; remove list of nemativororous species.

L. Is "awaited" the right word?

L. 132 square

L. 226, 228 randomized

L. 273 better describe this OTU: SH group in UNITE?

L. 348 plays

L. 364 lilacinum

L. 379 Calonectria

L. 534, 547, 549, 620 (

L. 695, 704 volume, pages missing

Reviewer: 2

Comments to the Author(s)

This paper dealt with relationship between microbial communities in rhizosphere and root of soybean plants and the infection by root-knot nematodes. Interestingly, authors sampling the soybean individual in the one field plot and separate the soybean individuals into three groups (normal, yellow and no leaf) corresponding to the infection of root-knot nematodes. Overall, the manuscript is well-written, methods and results are well presented and conclusions are fully justified.

However, the significantly weaknesses are that there were no any data on the nematode infections. It is quite easy to measure the root-knot index, which is quite necessary to explain how serious of the plants infected by nematode. Moreover, the infection of the nematodes (secondary-stage juveniles) occurred in the seedling stage of the plants. However, sampling was conducted at the late stage of the soybean. It is quite hard to correlate the microbes in the rhizosphere soil and roots to the nematode damaging. Although there are some relationships between some nematophagous microbes previously reported and nematode damaging, most of those microbes could survive as saprophytes in the soil. The other key issue is that root-knot nematode is kind of animals and many microbes can colonize on the different stages of root-knot nematode. However, there was no any examination on the microbial community on nematode. Following are some recent papers that may have reference value to help promote the significance of this manuscript:

The analysis of rhizosphere microbial communities during an invasion by *Ralstonia solanacearum* (Wei et al., *Ralstonia solanacearum* pathogen disrupts bacterial rhizosphere microbiome during an invasion, 2018), besides, the results on rhizosphere, root and cyst microbial communities in disease suppressive soil provide more insights into consortia of anti-nematode bacteria (Hussain et al., *Bacterial community assemblages in the rhizosphere soil, root endosphere and cyst of soybean cyst nematode-suppressive soil challenged with nematodes*, 2018).

Reviewer: 3

Comments to the Author(s)

I thought the overall science was sound, and the paper provides an interesting insight into the correlation between plant health, root-knot nematode colonization, and corresponding recruitment of nematophagous fungi and nematocidal bacteria to the root and rhizosphere. The paper makes a sound contribution to the literature on these complex interactions, and the authors are careful not to overinterpret their results. The only issue I found with the article was some minor formatting issues and some substantive grammatical concerns especially in the Summary, Introduction, and Discussion sections. These concerns are described in the attached review file.

Author's Response to Decision Letter for (RSOS-181693.R0)

See Appendix B.

RSOS-181693.R1 (Revision)

Review form: Reviewer 2

Is the manuscript scientifically sound in its present form?

Yes

Are the interpretations and conclusions justified by the results?

Yes

Is the language acceptable?

Yes

Is it clear how to access all supporting data?

Yes

Do you have any ethical concerns with this paper?

No

Have you any concerns about statistical analyses in this paper?

I do not feel qualified to assess the statistics

Recommendation?

Accept with minor revision (please list in comments)

Comments to the Author(s)

The manuscript is acceptable except a minor comment. Actually *Dactylellina* is trapping fungus and can capture secondary-stage juveniles and *Clonostachys*, *Pochonia* and *Purpureocillium* can parasitize on nematode eggs. Those fungi associated with no-leaf individuals of soybean, that means high nematode densities in no-leaf individuals can stimulate the multiply of those fungi. Authors may discuss this point a little bit.

Decision letter (RSOS-181693.R1)

07-Feb-2019

Dear Dr Toju:

On behalf of the Editors, I am pleased to inform you that your Manuscript RSOS-181693.R1 entitled "Consortia of anti-nematode fungi and bacteria in the rhizosphere of soybean plants attacked by root-knot nematodes" has been accepted for publication in Royal Society Open Science subject to minor revision in accordance with the referee suggestions. Please find the referees' comments at the end of this email.

The reviewers and Subject Editor have recommended publication, but also suggest some minor revisions to your manuscript. Therefore, I invite you to respond to the comments and revise your manuscript.

- Ethics statement

- Data accessibility

It is a condition of publication that all supporting data are made available either as supplementary information or preferably in a suitable permanent repository. The data accessibility section should state where the article's supporting data can be accessed. This section should also include details, where possible of where to access other relevant research materials such as statistical tools, protocols, software etc can be accessed. If the data has been deposited in

an external repository this section should list the database, accession number and link to the DOI for all data from the article that has been made publicly available. Data sets that have been deposited in an external repository and have a DOI should also be appropriately cited in the manuscript and included in the reference list.

<http://datadryad.org/submit?journalID=RSOS&manu=RSOS-181693.R1>

- **Competing interests**

- **Authors' contributions**

- **Acknowledgements**

- **Funding statement**

Because the schedule for publication is very tight, it is a condition of publication that you submit the revised version of your manuscript before 16-Feb-2019. Please note that the revision deadline will expire at 00.00am on this date. If you do not think you will be able to meet this date please let me know immediately.

on behalf of Dr Berat Haznedaroglu (Associate Editor) and Professor Kevin Padian (Subject Editor)
openscience@royalsociety.org

Reviewer comments to Author:
Reviewer: 2

Comments to the Author(s)

The manuscript is acceptable except a minor comment. Actually *Dactylellina* is trapping fungus and can capture secondary-stage juveniles and *Clonostachys*, *Pochonia* and *Purpureocillium* can parasitize on nematode eggs. Those fungi associated with no-leaf individuals of soybean, that

means high nematode densities in no-leaf individuals can stimulate the multiply of those fungi. Authors may discuss this point a little bit.

Author's Response to Decision Letter for (RSOS-181693.R1)

See Appendix C.

Decision letter (RSOS-181693.R2)

21-Feb-2019

Dear Dr Toju,

I am pleased to inform you that your manuscript entitled "Consortia of anti-nematode fungi and bacteria in the rhizosphere of soybean plants attacked by root-knot nematodes" is now accepted for publication in Royal Society Open Science.

on behalf of Dr Berat Haznedaroglu (Associate Editor) and Professor Kevin Padian (Subject Editor)
openscience@royalsociety.org

Appendix A

Review for RSOS root-knot nematode paper

Formatting concerns.

- 1) Figure and Table notations should be consistent throughout – I would suggest Bold, written out. For supplementary data/figures, these can just be labeled “Figure Sx” rather than redundantly labeling them “electronic supplementary material, Figure Sx”
- 2) The sections labeled Ethics, Data Accessibility, etc. These should be bolded
- 3) Tables should be rotated so the text can be big enough to read. Tables 2-4. Alternatively just use the most specific classification for the identified OTU (perhaps labeled with (P/O/F/C/G to clarify), to make the table easier to read. Also separate tables for Green and No Leaf associated OTUs would be clearer
- 4) The primers used for sequencing and amplification should be put in a Table, and the Tables in the paper should be renumbered.

Writing concerns.

The writing is generally clear, but there are a few bad habits that make for difficult reading. The most pervasive is the use of words like “However”, “Overall”, and “Therefore” to start sentences. In most cases these words can be removed without changing the meaning of the text. There are also several instances of run on sentences using several commas to delineate clauses that could be broken up into separate sentences. This is largely in the Summary, Introduction, and to a lesser extent in the discussion. Almost everywhere in the text where a sentence starts with a short clause followed by a comma, this clause can be removed or placed at the end of the sentence to make the writing clearer.

A specific concern in the discussion is that the preference analysis used to suggest that certain microbes are preferentially found on the roots of the diseased plants, but no quantitative data is provided (in other words, how much more prevalent is *Pseudomonas* in the “no leaf” rhizosphere?). Similar analysis on each of the OTUs found to have a preference could shed additional light on the magnitude of the effect of disease.

The discussion is pretty good, but a little long. The network analysis is discussed a lot, and I think lengthy discussion of the nematicidal properties of various organisms can be cut, considering that you don’t know if these specific organisms are present (only organisms in the same genus).

Appendix B

[revised manuscript text omitted]

Formatted: Font: Italic

**Ethics.** The fieldwork and sampling of materials were permitted by Crop Science Laboratory,
Graduate School of Agriculture, Kyoto University. No ethical assessment was required prior
to conducting this research. As this research does not target humans and animals, neither
informed consent nor animal ethical investigations were required.

Formatted: Font: Bold

**Data accessibility.** Data are available from the electric supplementary material, data S1-S5
and DNA Data Bank of Japan (DDBJ) (DRA006845).

Formatted: Font: Bold

**Authors' contributions.** H-T ~~conceived and~~ designed the work. H-T- and Y-T- performed
fieldwork. H-T. conducted molecular experiment and analyzed the data. H-T- wrote the
manuscript with Y-T-. All authors gave final approval for publication.

Formatted: Font: Bold

**Competing interests.** The authors declare no competing interests.

Formatted: Font: Bold

**Acknowledgements.** We thank Tatsuhiko Shiraiwa for his support in fieldwork and Mizuki
Shinoda, Ko Mizushima, Sarasa Amma, and Hiroki Kawai for their support in molecular
experiments. We are also grateful to Ryoji Shinya for his advice on the biology of nematodes.
We are also grateful to anonymous reviewers for their productive comments.

**Funding.** This work was financially supported by JSPS KAKENHI Grant (15KT0032) and
JST PRESTO (JPMJPR16Q6) to HT.

Formatted: Font: Bold

~~**Acknowledgements.** We thank Tatsuhiko Shiraiwa for his support in fieldwork and Mizuki~~
~~Shinoda, Ko Mizushima, Sarasa Amma, and Hiroki Kawai for their support in molecular~~
~~experiments. We are also grateful to Ryoji Shinya for his advice on the biology of nematodes.~~

Formatted: Font: Bold

**References**

- [1] Barker, K.R. & Koenning, S.R. 1998 Developing sustainable systems for nematode
management. *Ann. Rev. Phytopathol.* **36**, 165-205. (doi:10.1146/annurev.phyto.36.1.165).
- [2] Abad, P., Gouzy, J., Aury, J.-M., Castagnone-Sereno, P., Danchin, E.G., Deleury, E.,
Perfus-Barbeoch, L., Anthouard, V., Artiguenave, F. & Blok, V.C. 2008 Genome sequence of

the metazoan plant-parasitic nematode *Meloidogyne incognita*. *Nat. Biotech.* **26**, 909.
(doi:10.1038/nbt.1482).

[3] Wrather, J.A., Anderson, T., Arsyad, D., Gai, J., Ploper, L., Porta-Puglia, A., Ram, H. &
Yorinori, J. 1997 Soybean disease loss estimates for the top 10 soybean producing countries
in 1994. *Plant disease* **81**, 107-110. (doi:10.1094/PDIS.1997.81.1.107).

[4] Wrather, J.A. & Koenning, S.R. 2006 Estimates of disease effects on soybean yields in the
United States 2003 to 2005. *J. Nematol.* **38**, 173-180.

[5] Li, J., Zou, C., Xu, J., Ji, X., Niu, X., Yang, J., Huang, X. & Zhang, K.-Q. 2015 Molecular
mechanisms of nematode-nematophagous microbe interactions: basis for biological control of
plant-parasitic nematodes. *Ann. Rev. Phytopathol.* **53**, 67-95. (doi:10.1146/annurev-phyto-
080614-120336).

[6] Schmitt, D., Corbin, F. & Nelson, L. 1983 Population dynamics of *Heterodera glycines*
and soybean response in soils treated with selected nematicides and herbicides. *J. Nematol.*
**15**, 432-437.

[7] Meyer, S.L. & Roberts, D.P. 2002 Combinations of biocontrol agents for management of
plant-parasitic nematodes and soilborne plant-pathogenic fungi. *J. Nematol.* **34**, 1-8.

[8] Nusbaum, C. & Ferris, H. 1973 The role of cropping systems in nematode population
management. *Ann. Rev. Phytopathol.* **11**, 423-440.
(doi:10.1146/annurev.py.11.090173.002231).

[9] Okada, H. & Harada, H. 2007 Effects of tillage and fertilizer on nematode communities in
a Japanese soybean field. *Appl. Soil Ecol.* **35**, 582-598. (doi:10.1016/j.apsoil.2006.09.008).

[10] Thomas, S. 1978 Population densities of nematodes under seven tillage regimes. *J.*
*Nematol.* **10**, 24.

[11] Donald, P., Tyler, D. & Boykin, D. 2009 Short-and long-term tillage effects on
*Heterodera glycines* reproduction in soybean monoculture in west Tennessee. *Soil Tillage Res.*
**104**, 126-133. (doi:10.1016/j.still.2009.02.002).

[12] Hamid, M.I., Hussain, M., Wu, Y., Zhang, X., Xiang, M. & Liu, X. 2017 Successive
soybean-monoculture cropping assembles rhizosphere microbial communities for the soil
suppression of soybean cyst nematode. *FEMS Microbiol. Ecol.* **93**.

(doi:10.1093/femsec/fiw222).

[13] Edwards, J., Johnson, C., Santos-Medellín, C., Lurie, E., Podishetty, N.K., Bhatnagar, S.,
Eisen, J.A. & Sundaresan, V. 2015 Structure, variation, and assembly of the root-associated
microbiomes of rice. *Proc. Natl. Acad. Sci. USA*. **112**, E911-E920.
(doi:10.1073/pnas.1414592112).

[14] Lundberg, D.S., Lebeis, S.L., Paredes, S.H., Yourstone, S., Gehring, J., Malfatti, S.,
Tremblay, J., Engelbrektson, A., Kunin, V. & Del Rio, T.G. 2012 Defining the core
*Arabidopsis thaliana* root microbiome. *Nature* **488**, 86-90. (doi:10.1038/nature11237).

[15] Toju, H., Peay, K.G., Yamamichi, M., Narisawa, K., Hiruma, K., Naito, K., Fukuda, S.,
Ushio, M., Nakaoka, S., Onoda, Y., et al. 2018 Core microbiomes for sustainable
agroecosystems. *Nat. Plants* **4**, 247–257. (doi:10.1038/s41477-018-0139-4).

[16] Schlaeppi, K. & Bulgarelli, D. 2015 The plant microbiome at work. *Mol. Plant-Microbe*
*Int.* **28**, 212-217. (doi:10.1094/MPMI-10-14-0334-FI).

[17] Mendes, R., Kruijt, M., De Bruijn, I., Dekkers, E., van der Voort, M., Schneider, J.H.,
Piceno, Y.M., DeSantis, T.Z., Andersen, G.L. & Bakker, P.A. 2011 Deciphering the
rhizosphere microbiome for disease-suppressive bacteria. *Science* **332**, 1097-1100.
(doi:10.1126/science.1203980).

[18] Berendsen, R.L., Pieterse, C.M. & Bakker, P.A. 2012 The rhizosphere microbiome and
plant health. *Trends Plant Sci.* **17**, 478-486. (doi:10.1016/j.tplants.2012.04.001).

[19] Mendes, R., Garbeva, P. & Raaijmakers, J.M. 2013 The rhizosphere microbiome:
significance of plant beneficial, plant pathogenic, and human pathogenic microorganisms.
*FEMS Microbiol. Rev.* **37**, 634-663. (doi:10.1111/1574-6976.12028).

[20] Cha, J.-Y., Han, S., Hong, H.-J., Cho, H., Kim, D., Kwon, Y., Kwon, S.-K., Crüsemann,
515 M., Lee, Y.B. & Kim, J.F. 2016 Microbial and biochemical basis of a *Fusarium* wilt-
516 suppressive soil. *ISME J.* **10**, 119. (doi:10.1038/ismej.2015.95).

[21] Hussain, M., Hamid, M.I., Tian, J., Hu, J., Zhang, X., Chen, J., Xiang, M. & Liu, X. 2018
Bacterial community assemblages in the rhizosphere soil, root endosphere and cyst of
soybean cyst nematode-suppressive soil challenged with nematodes. *FEMS Microbiol. Ecol.*
**94**, fiy142.

[22] Gavassoni, W.L., Tylka, G.L. & Munkvold, G.P. 2001 Relationships between tillage and
spatial patterns of *Heterodera glycines*. *Phytopathology* **91**, 534-545.
(doi:10.1094/PHYTO.2001.91.6.534.).

[23] Sato, H. & Murakami, N. 2008 Reproductive isolation among cryptic species in the
ectomycorrhizal genus *Strobilomyces*: population-level CAPS marker-based genetic analysis.
*Mol. Phyl. Evol.* **48**, 326-334. (doi:10.1016/j.ympev.2008.01.033).

[24] Takada-Hoshino, Y. & Matsumoto, N. 2004 An improved DNA extraction method using
skim milk from soils that strongly adsorb DNA. *Microbes Env.* **19**, 13-19.
(doi:10.1264/jsme2.19.13).

[25] Caporaso, J.G., Lauber, C.L., Walters, W.A., Berg-Lyons, D., Lozupone, C.A.,
Turnbaugh, P.J., Fierer, N. & Knight, R. 2011 Global patterns of 16S rRNA diversity at a
depth of millions of sequences per sample. *Proc. Natl. Acad. Sci. USA.* **108**, 4516-4522.
(doi:10.1073/pnas.1000080107).

[26] Lundberg, D.S., Yourstone, S., Mieczkowski, P., Jones, C.D. & Dangl, J.L. 2013
Practical innovations for high-throughput amplicon sequencing. *Nat. Methods* **10**, 999-1002.
(doi:10.1038/nmeth.2634).

[27] Apprill, A., McNally, S., Parsons, R. & Weber, L. 2015 Minor revision to V4 region SSU
rRNA 806R gene primer greatly increases detection of SAR11 bacterioplankton. *Aquat*
*Microb Ecol* **75**, 129-137. (doi:10.3354/ame01753).

[28] Stevens, J.L., Jackson, R.L. & Olson, J.B. 2013 Slowing PCR ramp speed reduces
chimera formation from environmental samples. *J. Microbiol. Methods* **93**, 203-205.
(doi:10.1016/j.mimet.2013.03.013).

[29] Hamady, M., Walker, J.J., Harris, J.K., Gold, N.J. & Knight, R. 2008 Error-correcting
barcoded primers for pyrosequencing hundreds of samples in multiplex. *Nat. Methods* **5**, 235-
237. (doi:10.1038/nmeth.1184).

[30] Toju, H., Tanabe, A.S., Yamamoto, S. & Sato, H. 2012 High-coverage ITS primers for
the DNA-based identification of ascomycetes and basidiomycetes in environmental samples.
*PLOS ONE* **7**, e40863. (doi:10.1371/journal.pone.0040863).

[31] Tanabe, A.S. 2018 Claident v0.2.2018.05.29, a software distributed by author at
<http://www.fifthdimension.jp/>.

[32] Tanabe, A.S. & Toju, H. 2013 Two new computational methods for universal DNA
barcoding: a benchmark using barcode sequences of bacteria, archaea, animals, fungi, and
land plants. *PLoS ONE* **8**, e76910. (doi:10.1371/journal.pone.0076910).

[33] Rognes, T., Mahé, F., Flouri, T., Quince, C. & Nichols, B. 2014 Vsearch: program
available at <https://github.com/torognes/vsearch>.

[34] Huson, D.H., Auch, A.F., Qi, J. & Schuster, S.C. 2007 MEGAN analysis of metagenomic
data. *Genome Res.* **17**, 377-386. (doi:10.1101/gr.5969107).

[35] Toju, H., Tanabe, A. & Ishii, H. 2016 Ericaceous plant–fungus network in a harsh alpine–
subalpine environment. *Mol. Ecol.* **25**, 3242-3257. (doi:10.1111/mec.13680).

[36] Toju, H., Yamamoto, S., Tanabe, A.S., Hayakawa, T. & Ishii, H.S. 2016 Network
modules and hubs in plant-root fungal biome. *J. R. Soc. Interface* **13**, 20151097.
(doi:10.1098/rsif.2015.1097).

[37] Peay, K.G., Russo, S.E., McGuire, K.L., Lim, Z., Chan, J.P., Tan, S. & Davies, S.J. 2015
Lack of host specificity leads to independent assortment of dipterocarps and ectomycorrhizal
fungi across a soil fertility gradient. *Ecol. Lett.* **18**, 807-816. (doi:10.1111/ele.12459).

[38] Oksanen, J., Blanchet, F.G., Kindt, R., Legendre, P., Minchin, P.R., O'Hara, R.B.,
Simpson, G.L., Solymos, P., Stevens, M.H.H. & Wagner, H. 2012 Vegan: community ecology
package. R package version 2.0-3 available at <http://CRAN.R-project.org/package=vegan>.

[39] R-Core-Team. 2018 R 3.5.1: A language and environment for statistical computing
available at <http://www.R-project.org/>.

[40] Anderson, M.J. 2001 A new method for non-parametric multivariate analysis of variance.
*Austral Ecol.* **26**, 32-46. (doi:10.1111/j.1442-9993.2001.01070.pp.x).

[41] Kurtz, Z.D., Mueller, C.L., Miraldi, E.R., Littman, D.R., Blaser, M.J. & Bonneau, R.A.
2015 Sparse and compositionally robust inference of microbial ecological networks. *PLoS*
*Comp. Biol.* **11**, e1004226. (doi:10.1371/journal.pcbi.1004226).

[42] Csardi, G. & Nepusz, T. 2006 The igraph software package for complex network
research. *Int. J. Complex Syst.* **1695**, 1-9.

[43] Esnard, J., Potter, T.L. & Zuckerman, B.M. 1995 *Streptomyces costaricanus* sp. nov.,
isolated from nematode-suppressive soil. *International Journal of Systematic and*

*Evolutionary Microbiology* **45**, 775-779. (doi:10.1099/00207713-45-4-775).

[revised manuscript text omitted]

Appendix C

Dear Dr. Haznedaroglu and Prof. Padian,

We would like to re-submit our manuscript entitled “Consortia of anti-nematode fungi and bacteria in the rhizosphere of soybean plants attacked by root-knot nematodes” (RSOS-181693.R1; bioRxiv accession, <http://biorxiv.org/cgi/content/short/365023v1>) for possible publication in *Royal Society Open Science*.

We appreciate the reviewer for his/her constructive comments. Responses to the comments are shown below

This manuscript has never been published before and is not currently being considered for publication elsewhere. The manuscript has been deposited on the bioRxiv preprint server (doi: <https://doi.org/10.1101/332403>). We confirm that the manuscript has been read and approved by all authors.

We hope that we have addressed reviewer comments adequately and constructively.

Sincerely,

Hirokazu Toju

Center for Ecological Research, Kyoto University, Hirano 2-509-3, Otsu, Shiga
520-2113, Japan

E-mail: toju.hirokazu.4c@kyoto-u.ac.jp

Tel.: +81-77-549-8234

Fax.: +81-77-549-8201

Reviewer: 2

Comments to the Author(s)

The manuscript is acceptable except a minor comment. Actually *Dactylellina* is trapping fungus and can capture secondary-stage juveniles and *Clonostachys*, *Pochonia* and *Purpureocillium* can parasitize on nematode eggs. Those fungi associated with no-leaf individuals of soybean, that means high nematode densities in no-leaf individuals can stimulate the multiply of those fungi. Authors may discuss this point a little bit.

Response:

The suggested information of *Dactylellina* and other anti-nematode fungi has been included in the revised manuscript (1.353-355, 359-378).